# Body image and appearance distress among military veterans and civilians with an injury-related visible difference: A comparison study

**Mary Keeling** [1¤a]*, **Diana Harcourt** [1], **Paul White** [2], **Sarah Evans** [1], **Victoria S. Williams V.** [1¤b], **James Kiff** [1¤c], **Heidi Williamson** [1]

**1** Centre for Appearance Research, School of Social Sciences, College of Health, Science and Society, University of the West of England, Frenchay Campus, Bristol, England, **2** Department of Mathematics and Statistics, University of the West of England, Bristol, England

¤a Current address: Defence and Security, RAND Europe, Cambridge, United Kingdom
¤b Current address: North Bristol NHS Trust, Bristol, United Kingdom
¤c Current address: Outlook Team, Southmead Hospital, North Bristol NHS Trust, Bristol, England
* mkeeling@randeurope.org

## Abstract

Injuries sustained during military conflict can significantly impact appearance. Yet, little is known about the psychosocial experiences of veterans with conflict-related appearance-altering injuries (AAI) and whether current civilian interventions are appropriate for this group. To inform the development of acceptable and effective support for veterans with appearance–related psychosocial difficulties, this study aimed to identify factors associated with psychosocial adjustment to an altered appearance among both veterans and civilians with AAI. A cross-sectional online survey was completed by 121 veterans and 197 civilians who had sustained AAI. Multivariable regression was used to examine factors related to adjustment in the two groups. Overall, both groups reported similar experiences, with some key exceptions. Veterans reported significantly greater depression and Post Traumatic Stress Disorder, significantly lower Body Image (BI) psychological flexibility, BI life engagement, and higher perceived appearance-related stigma. BI psychological flexibility was identified as a key predictor of appearance-related outcomes in both groups. Self-compassion predicted social anxiety and depression symptoms in both groups, but only appearance outcomes among civilians. Based on these identified associated factors, it is suggested that both groups, but particularly veterans, may benefit from an Acceptance and Commitment Therapy-based intervention, including explicit self-compassion activities, and practical social skills training.

## Introduction

Military conflict injuries can profoundly affect appearance. UK Ministry of Defence data shows that between April 2005 and March 2020, 10,100 current and former military personnel received compensation due to 'injury, wounds, and scarring' (excluding musculoskeletal

**Data Availability Statement:** The relevant data may not be shared publicly as it contains sensitive and potentially identifying information about participants' appearance-altering injuries. In

addition, participants did not consent to their data being shared. Requests for data can be sent to the UNITS research team at the Centre for Appearance Research (car@uwe.ac.uk).

**Funding:** The Scar Free Foundation awarded funding to Heidi Williamson and Diana Harcourt for the conduct of this research. https://scarfree.org.uk/ The funders did not play a role in any aspects of the research other than funding.

**Competing interests:** The authors have declared that no competing interests exist.

injuries) [1], and 362 received traumatic or surgical amputations [1]. Despite substantial research on altered appearance (visible difference/disfigurement) from various causes, there has been limited focus on the unique experiences and support requirements of military personnel and veterans with appearance-altering injuries (AAI) from military conflict.

Outside the military context, looking different from the norm due to factors like burn injuries or limb loss can lead to enduring psychosocial challenges including negative effects on body image, self-esteem, and confidence [2, 3] and feelings of anger and hostility [3]. Common difficulties include coping with stigmatizing reactions from others such as staring, inappropriate comments, avoidance, and unsolicited questions [4], concerns around the impact on intimate relationships when disclosing their difference, and on employment [5]. While some individuals effectively navigate these challenges and report personal growth [4], others struggle, experiencing social avoidance and isolation in response to others' reactions and the fear of negative judgment [2].

Whilst there is considerable similarity in the challenges reported by people with an unusual appearance irrespective of its type or cause [6], understanding the factors influencing adjustment is crucial. Research with burns patients highlights the importance of paying particular attention to appearance-related issues when supporting people who have endured traumatic injuries (Shepherd, 2015 [7]). Among 1,265 non-military adults with diverse visible differences, evidence has shown the significance of psychosocial factors, rather than demographics or condition-related aspects [8]. Specifically, outlook on life (disposition) and feeling accepted and supported by others positively influenced outcomes including social anxiety and avoidance. Valence (the value attributed to appearance) and the importance (salience) of appearance in the self-concept may also contribute to adjustment [2]. A review of research with burns patients found that avoidant coping was associated with negative adjustment, and active or acceptance coping with positive adjustment [9]. Recently, psychological flexibility, characterized by the ability to stay present despite distressing thoughts, feelings, and bodily sensations, and aligning decisions with personal values [10], has been examined in relation to appearance-related distress amongst adults with burns [11] and other visible differences [12].

It's crucial not to assume that the experience of having an altered appearance, the factors affecting outcomes, and support requirements, are identical for individuals with military-related injuries compared to those without military backgrounds. To date, there has been limited exploration of the impact of being part of an organisation where physical prowess is key, and of the influence of military culture that values mental toughness and deplores signs of weakness.

A recent review [13] found only four papers specifically addressing body image and psychosocial issues among military personnel and veterans with AAI. Weaver et al. [14] discovered positive correlations between body image distress and depressive symptoms, with a trend towards body image distress being associated with symptoms of post-traumatic stress disorder (PTSD) among military veterans with AAI. Similarly, Akyol et al. [15] found body image distress was linked to self-reported depressive symptoms among 60 Turkish military personnel with lower limb amputations. Two case studies of US military male upper limb amputees [16] mentioned the impact of appearance concerns on social relationships. While most research in this area has primarily involved men, Cater [17] noted body image issues among six US servicewomen who experienced traumatic limb loss and described loss of confidence and concern about how they were viewed by the public, including challenges around meeting new people and dealing with hurtful comments. Factors including a positive attitude, social support, personal courage, resilience, humor, and military training and culture, positively influenced their recovery. Keeling et al.'s [13] review concluded that injured personnel and veterans with AAI can encounter psychosocial challenges akin to those experienced by civilians with visible

differences. However, additional factors, including military culture, may influence their adjustment and resilience to appearance-related challenges, impacting their support needs.

We previously interviewed 20 veterans and 3 serving personnel who had sustained AAI during deployment or training. While some experiences aligned with civilian evidence, these interviews revealed distinct military-related nuances in challenges, protective factors, coping strategies, and barriers to and preferences for support [18]. Notably, our interviews indicated that acceptance and adjustment could hinge on perceptions of injury likelihood and a 'hierarchy of injuries' where combat-related injuries were viewed as 'heroic.' Peer support, an optimistic disposition, compassion, and drawing comparisons with others' injuries were also influential. Participants wanted support with appearance-related challenges but expressed concerns that, in a military context, raising appearance-related issues might be seen as vanity. This underscores the existence of unmet support needs among UK military personnel and veterans with AAI, emphasizing the necessity to better understand adjustment in this group.

To date, no large-scale studies have compared people with AAI from military and civilian backgrounds, to determine the extent to which the psychosocial impact of their AAI and the factors influencing their adjustment are similar or different. To address this gap in the literature and to inform the provision of evidence-based psychosocial support to meet any specific needs of military veterans with AAI, this exploratory study aimed to answer the question, to what extent are the factors that predict psychosocial adjustment to an altered appearance among UK military veterans injured in a military conflict context similar or different to those that predict adjustment among a comparative sample of adults without a military background?

For this study, and based on the presented existing literature, adjustment was conceptualised as satisfaction with appearance (body esteem), the impact of appearance-related concerns on participation in social, recreational, and vocational activities (body image life engagement), the level of concern about being negatively judged based on appearance (fear of negative appearance evaluation), as well as social anxiety and depression. Factors believed to influence adjustment included body image psychological flexibility, self-compassion, engagement in meaningful activities, perceived social support, perceived appearance-related stigmatization, dispositional outlook (optimism), the use of appearance-fixing behaviors (efforts to conceal, alter, or avoid one's appearance), and symptoms of PTSD. Consequently, hypotheses of associations between appearance and psychological outcomes and the proposed predictor factors, for civilians and veterans, are:

Body Image psychological flexibility (a domain-specific version of psychological flexibility), self-compassion, engagement in meaningful activities, optimism and social support will be positively associated with body esteem and body image life engagement, and negatively associated with fear of negative appearance evaluation, social anxiety, and depression. Appearance fixing, perceived stigma and PTSD will be negatively associated with body esteem and body image life engagement, and positively associated with fear of negative appearance evaluation, social anxiety, and depression.

## Method

### Design

A cross-sectional survey was utilised including two participant groups, veterans (those who have left military service), and civilians with AAI.

### Sample and participants

Following Tabachnick & Fidell [19], based on the inclusion of eight validated predictor variables, we aimed to recruit 120 veterans who had sustained an AAI. This target was inflated to 200 to

allow non-validated predictor variables to probe the model. For comparability we additionally aimed for a target sample of 200 civilians with AAI who had never served in the military.

Participants were 121 veterans who had sustained an appearance-altering (as they perceive it) physical injury either during operational deployment as a result of enemy action (e.g., blast or gunshot) or an accident (e.g., a motor vehicle accident); or during field training in preparation for operational deployment. This must have been during active service in the UK Armed Forces any time since 1969 (to include those injured in 'The Troubles' in Northern Ireland, the Falklands conflict, and First Gulf War, as well as more recent conflicts). The injury must have occurred at least one year prior to participation, excluding those in acute medical recovery and rehabilitation. The civilian sample consisted of 197 adults with no military background who sustained an AAI (as they perceive it) such as a burn or limb loss, when aged 18 years or older, between 1969 and at least one year prior to data collection.

## Materials

Two surveys were created for data collection from veteran and civilian participants. The surveys were identical, except for veteran-specific questions covering military service details and experiences identified in a prior qualitative study as potential predictors of body image and appearance-related outcomes. The research team, drawing on expertise in visible differences, body image, and military health, and the results of their previous qualitative study [18], developed the surveys with input from Public and Patient Involvement (PPI) advisors, with relevant lived experiences. The surveys covered: Socio-demographics; Military career (veterans only); Injury-specific questions; Being a veteran (veterans only); Health and fitness; Appearance concerns; Social wellbeing; Mental health; Family and relationships; Support experiences, barriers to care and support preferences (veterans only).

## Procedure

After obtaining ethics approval from the University of the West of England, University Research Ethics Committee (Ref: HAS.19.12.086) and Health Research Authority (IRAS ID: 257931), participant recruitment followed a six-pronged approach: 1. Veteran participants from a prior qualitative study [18] were invited via email, having consented to future research contact. 2. Relevant veteran and civilian support organizations shared study information via their usual communication modes and/or shared study invites directly with eligible service users. 3. Relevant NHS services across England and Wales, including outpatients' clinics for burns and prosthetics services shared study adverts via their usual communication modes and/or shared study invites directly with eligible service users, and/or distributed paper surveys to individuals attending the service in-person. 4. Study adverts were shared on social media via Twitter (X), Instagram, and Facebook. 5. Study information was posted on Reddit. 6. Advertisements were placed in relevant veteran magazines.

Potential participants, upon seeing the study advert or receiving an invitation letter, accessed a secure online survey via Qualtrics. Emailed or paper versions of the survey were also made available for those preferring these mediums. Participant recruitment took place between March and November 2020. In both versions, participants provided written informed consent on the first page of the survey before progressing to the survey questions. Due to the recruitment approach, the exposure count for study invitations is unknown, rendering a response rate unavailable.

## Measures

**Outcome measures.** Six validated outcome measures (three appearance and three mental health) were included in both survey versions.

*Appearance outcome measures.* The Appearance Esteem subscale (10 items) of the Body Esteem Scale for Adults and Adolescents (**BESAA-AE**) [20] assesses general feelings and satisfaction with appearance (e.g., "I worry about the way I look") using a 5-point scale (never to always). Mean scores, indicating greater body esteem with higher values, were computed. The scale has demonstrated reliability and validity in adults and adolescents [20], and internal consistency in this study was high ($\alpha = 0.932$), consistent with prior research in adults with visible differences due to burn scarring ($\alpha = 0.95$) [21].

The Body Image Life Engagement Questionnaire (**BILEQ)** [22] measures the wider impact of appearance-related concerns on various life domains, focusing on behavioral avoidance due to negative feelings about one's appearance. Participants were asked, "In the past two weeks, how much have worries or feeling bad about the way you looked stopped you from doing any of the following things?" The 11-item scale covered social, recreational, and educational/vocational activities (e.g., "doing physical activity/sport") on a 4-point scale (1 = hasn't stopped me at all, 4 = stopped me all the time). All items were reverse-scored, and a mean was calculated. Higher scores denoted greater life engagement. The scale demonstrated good internal consistency ($\alpha = 0.924$), consistent with prior research with female adults [22].

The Fear of Negative Appearance Evaluation Scale (**FNAES**) [23] assesses participants' concern about others evaluating them negatively based on their appearance. Six statements (e.g., "I am concerned about what others think of my appearance") are rated on a 5-point scale (1 = Not at all, 5 = Extremely). Scores are summed with a higher score indicating greater fear of negative appearance evaluation. The scale has demonstrated construct and predictive validity in undergraduate college students [23] and good internal consistency in adults born with cleft lip and palate [24, 25]. In this study, internal consistency was high ($\alpha = 0.938$).

*Mental health outcome measures.* Two sub-scales from The Social Anxiety Scale for Adolescents (SAS-A) [25, 26] measured social anxiety: social avoidance and distress in new situations (**SAS-new**) and social avoidance and distress in general (**SAS-general**). SAS-new includes six statements (e.g., "I get nervous when I meet new people"), while SAS-general comprises four (e.g., "I feel shy even with peers I know very well"). Both are rated on a 5-point scale (1 = not at all, 5 = all the time), with higher scores indicating greater social anxiety. Satisfactory Cronbach's alphas have been reported for the subscales (0.83 for SAS-new and 0.76 for SAS-general) [25]. Internal consistency in the current study was $\alpha = 0.913$ for SAS-new and $\alpha = 0.872$ for SAS-general.

The Patient Health Questionnaire-9 (**PHQ-9)** [26] assesses depression symptoms across the nine Diagnostic and Statistical Manual criteria. Individuals rate the extent to which they have been bothered by each symptom over the past two weeks using a 4-point scale ('not at all' to 'nearly every day'). Scores range from 0 to 27, with 5–9 indicating 'mild depression' and a clinical cut point score of 10 or above indicating moderate to severe depression. The PHQ-9 demonstrates strong internal consistency (Cronbach's $\alpha = 0.89$) and test–retest reliability (r = 0.84) [26]. Internal consistency in the current study was $\alpha = 0.919$.

**Explanatory variables.** Explanatory variables included eight validated measures aligned with the study hypotheses and 12 non-validated items (created by the authors) identified as potentially associated with appearance and body image experiences in a previous qualitative study [18]. The eight validated measures were:

The Body Image Acceptance and Action Questionnaire-5 (**BIAAQ-5**; [27]) measures body image flexibility (fully experiencing perceptions, sensations, feelings, thoughts, and beliefs about the body while intentionally pursuing effective action in other life domains), a component of positive body image. It consists of five items, addressing both acceptance and action. All items are reverse-scored and summed, with higher scores indicating greater body image flexibility. This version has demonstrated good reliability and validity in a mixed-gender

sample of US adults [27]. Minor edits were made, focusing on changed appearance instead of weight and body shape. Internal consistency in the current study was α = 0.919.

The Appearance Fixing sub-scale of the Body Image Coping Strategies Inventory (**BICSI-AF**) [28] measures participants' tendencies to cover, camouflage, seek reassurance, and engage in social comparison regarding the aspect(s) of their appearance that concerns them. The 10-item sub-scale includes statements like "I make a special effort to hide or cover up what's troublesome about my looks," rated between 0 = definitely not like me and 3 = definitely like me. Higher mean scores indicate increased use of appearance-fixing coping strategies. It has demonstrated good internal consistency, construct, and convergent validity among college students [28] and good internal consistency in a sample of mixed-gender adults with a visible difference following head and neck cancer surgery [29]. Internal consistency in the current study was α = 0.907.

The Perceived Stigmatisation Questionnaire (**PSQ**) [21] measures stigmatisation behaviours commonly experienced by people with a visible difference. It comprises three subscales: 1. Absence of friendly behaviour, 2. Confused/staring behaviour, and 3. Hostile behaviour. A high score reflects high-perceived stigmatisation. The PSQ has demonstrated good internal consistency, and convergent and discriminant validity among a sample of mixed gender adult burn survivors [21]. Internal consistency in the current study was α = 0.899.

Self-compassion was measured using the 12-item Short Form of the Self Compassion Scale (**SCS-SF**) [30]. A mean score is calculated with higher scores indicating higher levels of self-compassion. The SCS-SF exhibits near perfect correlation with the full SCS and demonstrates good internal consistency with both Dutch and American university students [30]. For this study, minor edits were made to improve readability as our PPI advisors fed back that some items were difficult to understand. Internal consistency in the current study was $\alpha$ = 0.895.

The Engagement in Meaningful Activities Survey (**EMAS**) is a validated 12-item measure of positive subjective experiences associated with day-to-day activities such as meaningful occupations (α = .91; [31]. Internal consistency in the current study was $\alpha$ = 0.924.

The Life Orientation Test–Revised (**LOT-R**) is a 10-item measure of dispositional optimism which has been shown to possess adequate predictive and discriminant validity [32]. Internal consistency in the current study was $\alpha$ = 0.848.

Multidimensional Scale of Perceived Social Support (**MSPSS**) is a validated measure of subjectively assessed social support across three areas of family, friends, and significant others [33]. Internal consistency in the current study was $\alpha$ = 0.941.

The Posttraumatic Stress Disorder (PTSD) items from the International Trauma Questionnaire (**ITQ**) [34] consisted of six questions covering three symptom clusters: (1) re-experiencing in the here and now, (2) avoidance, and (3) sense of current threat. Additionally, three indicators of functional impairment associated with these symptoms were assessed. Respondents were asked to provide a brief description of 'the experience that troubles them the most,' and questions were answered in relation to that experience. A PTSD diagnosis requires endorsement of at least one symptom from each of the three clusters and at least one indicator of functional impairment. In addition to calculated PTSD case scores, summed PTSD scores were also calculated from individual item scores. The ITQ has been validated for use in the general population [34] and with treatment-seeking veterans [35]. Internal consistency in the current study was high (α = 0.952).

The 12 non-validated items included six veteran-specific items: 1. "Expecting to be injured" (six-point ordinal scale); 2. "Being seen as an injured veteran is a good thing" (five-point ordinal scale); 3. "Having other veterans close by who are recovering from similar injuries and changes to their appearance has been helpful" (five-point ordinal scale); 4. Which operational deployment they were injured on, e.g. Afghanistan; 5. "I would not feel comfortable talking about my altered appearance following my injury while in a military environment" (six-point

ordinal scale); 6. Whether injury was on deployment or training (binary); and six items common to both the veteran and civilian sample: 7. Number of years since injury; 8. "Being physically fit is important to me" (six-point ordinal scale); 9. 'I tend to say to myself "it could have been worse‴' (Binary); 10. "I feel disgust when I think about or look at my appearance/body" (five-point ordinal scale); 11. "I use humor to defuse awkward conversations about my injury and/or altered appearance" (4-point ordinal scale); 12. "How noticeable do you think your scars/limb loss are to other people when you are fully clothed?" (11-point ordinal scale).

## Method of analysis

Each of the six outcome measures was compared between the veteran sample and the civilian sample using the separate variances independent samples t-test (Welch test) and effect size quantified using Hedge's g (with $0.2 <= g < 0.5$, $0.5 <= g < 0.8$, and $g >= 0.8$ being indicative of a small, medium or large effect respectively).

Multivariable least squares regression was used to relate the validated explanatory variables to outcome variables in both samples separately, and in all cases underlying statistical assumptions for valid inference were examined with the plan of taking remedial action should the need arise. Explanatory variables formed a mutually correlated system in both the veteran and civilian sample however the extent of multicollinearity, measured by the variance inflation factors (VIFs), were not a cause for concern for model building and interpretation (maximum VIF in the veteran population 3.33; maximum VIF in the civilian population 3.32). The maximum absolute value for the extent of skew in model residuals in any model did not cast doubt on appropriateness of model (maximum skew = 0.72) however, one model in the civilian sample indicated a large degree of excess kurtosis (4.02) consistent with the presence of potential outliers. Inspection of Normal quantile-quantile plots indicated the presence of two outliers in the model with the highest kurtosis, but otherwise all other normal quantile-quantile plots did not cause doubt on model appropriateness. A sensitivity analysis undertaken by temporarily deleting the two ill-fitting observations left statistical conclusions unchanged, and as such models using all available data are reported.

Multivariable regression is based on complete cases. N = 113 out of a sample of 121 veterans gave complete case data (i.e., 7% missing data in the veteran sample for at least one regression analysis). N = 161 out of a sample of 198 civilians gave complete case data (i.e., 19% missing data for at least one regression analysis). Little's MCAR tests was consistent with data missing completely at random ($\chi^2$ = 77.920, df = 100, p = 0.950). Full specification multiple imputation chained equations (MICE) was used with 100 imputations. The pooled regression results using MICE did not materially alter statistical conclusions except in two marginal cases. We therefore report the available case regression and additionally report the multiple imputed regression summaries in supplementary material.

Each of the six regression models were further probed in an exploratory manner using the 12 non-validated measures to determine if their inclusion would significantly improve the model and hence give an insight to additional factors affecting adjustment.

All statistical analysis was conducted using IMB SPSS for Windows, Version 24.0.

## Results

Table 1 summarises demographic and injury information of the participants. Veterans (median age = 42 years, age range 28 to 75 years) were typically older than civilians (median age = 35 years, age range 18 to 80) with the veteran sample being predominantly male (93.4%) and married (63.6%) compared with the civilian sample (60.1% male; 35.4% married). Both samples were predominantly of white ethnicity (96.7% veteran sample: 87.4% civilian sample). A higher proportion of civilians held graduate (37.8%) and postgraduate (28.7%) qualifications

**Table 1. Demographics of sample.**

| Gender | Veterans n (%) | Civilians n (%) |
|---|---|---|
| Male | 113 (93.4) | 119 (60.1) |
| Female | 8 (6.6) | 77(38.9) |
| Prefer to self-describe | 0 (0.0) | 1 (0.5) |
| Prefer not to say | 0 (0.0) | 1 (0.5) |
| **Ethnicity** | | |
| Asian or Asian British | 0 (0.0) | 5 (2.5) |
| Black, African, Black British or Caribbean | 1 (0.8) | 3 (1.5) |
| Mixed or multiple ethnicities | 0 (0.0) | 9 (4.5) |
| White | 117 (96.7) | 173 (87.4) |
| Other ethnicity | 0 (0.0) | 4 (2.0) |
| Prefer not to say | 3 (2.5) | 4 (2.0) |
| **Relationship Status** | | |
| Single | 14 (11.6) | 54 (27.3) |
| Relationship less than 6-months | 3 (2.5) | 4 (2.0) |
| Relationship more than 6-months | 1 (0.8) | 23 (11.6) |
| Living with partner | 13 (10.7) | 37 (18.7) |
| Married | 77 (63.6) | 70 (35.4) |
| Separated | 8 (6.6) | 3 (1.5) |
| Divorced | 5 (4.1) | 4 (2.0) |
| Widowed | 0 (0.0) | 3 (1.5) |
| **Education** | | |
| GCSEs or less | 42 (36.8) | 30 (16.0) |
| A-levels or equivalent | 29 (25.4) | 33 (17.6) |
| Degree or equivalent | 25 (21.9) | 71 (37.8) |
| Postgraduate qualifications | 18 (15.8) | 54 (28.7) |
| **Type of injury** | | |
| Both limb-loss and scarring | 64 (52.9) | 26 (13.3) |
| Limb-loss | 1 (0.8) | 0 (0.0) |
| Misshapen body part | 0 (0.0) | 2 (1.0) |
| Scars | 56 (46.3) | 170 (85.9) |
| **Cause of injury** | | |
| Accident on deployment | 23 (19.0) | - |
| Enemy action on deployment | 73 (60.3) | - |
| Training accident | 25 (20.7) | - |
| Accident/Explosion | - | 29 (14.6) |
| Assault/Violent crime | - | 16 (8.1) |
| Burns | - | 31 (15.7) |
| Road traffic accident/Transport | - | 64 (32.3) |
| Sport-related | - | 58 (29.3) |
| **Years since injury** (mean; SD) | 18.11 (11.68) | 8.02 (9.49) |
| Range | 2–32 | 1–48 |
| ***Military characteristics*** | | |
| **Service Branch** | | - |
| Naval Services | 16 (13.2) | - |
| Army | 100 (82.6) | - |
| Royal Air Force | 5 (4.1) | - |
| **Rank** | | - |

(*Continued*)

**Table 1.** (Continued)

| Gender | Veterans n (%) | Civilians n (%) |
|---|---|---|
| Other ranks | 42 (34.7) | - |
| NCO | 58 (47.9) | - |
| Officer | 21 (17.4) | - |
| **Engagement type** | | - |
| Regular | 117 (98.3) | - |
| Reserve | 2 (1.7) | - |
| **Years served in the military** (mean; SD) | 13.31 (7.62) | - |
| Range | 2–32 | - |
| *Years served when injured* (mean; SD) | 7.53 (6.08) | - |
| Range | 0–28 | - |

compared to veterans (21.9% graduate, 15.8% postgraduate). Civilians predominantly had scarring as their visible difference (85.9%), whereas veterans had a mix of scarring and limb loss (52.9%) or just scarring (46.3%). Sixty percent of veterans had been injured during enemy action. A third of the civilians were injured in road traffic or other transport-related accidents (32.3%), and just under a third in sport-related incidents (28.3%). Veterans on average had more years since their injury (18.11 mean years) compared to the civilians (8.02 mean years). Most veterans had served in the Army (82.6%) and were in the regular force at the time of their injury (98.3%). Among veterans, 47.9% held non-commissioned officer (NCO) rank, and 34.7% held ranks lower than NCO at the time of their injury.

Table 2 summarises means and standard deviations for each validated measure for the veteran and civilian samples. The statistical comparison for mean differences is in Table 3. There are significantly lower sample means for veterans on the Body Image Life Engagement Questionnaire (**BILEQ;** $p < .001$, $g = 0.64$) and on the Body Image Acceptance and Action Questionnaire (**BIAAQ;** $p < .001$, $g = 0.46$). There are significantly higher sample means for veterans on the Patient Health Questionniare-9 (**PHQ**-9; $p = .007$, $g = 0.35$), the Body Image Coping Strategies Inventory–Appearance Fixing (**BICSI-AF**; $p = .036$, $g = 0.25$), the Perceived

**Table 2. Mean score and standard deviation of all measures for veteran and civilian samples.**

| Measure | Veteran | | | Civilian | | |
|---|---|---|---|---|---|---|
| | N | Mean | SD | N | Mean | SD |
| BESAA-AE | 121 | 2.23 | 0.988 | 198 | 2.42 | 0.913 |
| BILEQ | 118 | 3.40 | 0.684 | 191 | 3.76 | 0.465 |
| FNAES | 118 | 13.28 | 6.674 | 189 | 14.03 | 6.607 |
| SAS-new | 113 | 15.40 | 6.749 | 175 | 16.89 | 6.418 |
| SAS-general | 113 | 9.24 | 4.175 | 175 | 9.54 | 3.956 |
| PHQ-9 | 113 | 9.35 | 7.457 | 174 | 7.05 | 6.004 |
| BIAAQ | 120 | 26.50 | 7.863 | 194 | 29.64 | 6.668 |
| BICSI-AF | 118 | 0.75 | 0.646 | 188 | 0.92 | 0.705 |
| PSQ | 114 | 2.14 | 0.516 | 180 | 1.89 | 0.479 |
| SCS-SF | 113 | 2.89 | 0.822 | 176 | 3.03 | 0.863 |
| EMAS | 114 | 33.66 | 6.915 | 175 | 33.38 | 7.610 |
| LOT-R | 113 | 12.25 | 5.336 | 174 | 12.89 | 5.117 |
| MSPSS | 110 | 58.61 | 18.358 | 169 | 61.40 | 16.421 |
| ITQ | 110 | 10.13 | 7.951 | 161 | 7.12 | 6.676 |

**Table 3. Mean difference between veteran and civilian scores on all variables.**

| Measure | Mean Difference | 95% CI | p | Hedge's g | 95% CI |
|---|---|---|---|---|---|
| BESAA-AE | -0.189 | -0.408, 0.029 | .089 | -0.20 | -0.34, 0.03 |
| BILEQ | -0.361 | -0.501, -0.220 | < .001 | -0.64 | -0.88, -0.41 |
| FNAES | -0.752 | -2.289, 0.784 | .336 | -0.11 | -0.34, 0.12 |
| SAS-new | -1.487 | -3.062, 0.087 | .064 | -0.23 | -0.46, 0.01 |
| SAS-general | -0.304 | -1.277, 0.669 | .539 | -0.07 | -0.31, 0.16 |
| PHQ-9 | 2.293 | 0.645, 3.942 | .007 | 0.35 | 0.11, 0.59 |
| BIAAQ | -3.139 | -4.840, -1.439 | < .001 | -0.44 | -0.67, -0.21 |
| BICSI-AF | -0.166 | -0.320, -0.011 | .036 | -0.25 | -0.48, -0.02 |
| PSQ | 0.256 | 0.137, 0.374 | < .001 | 0.51 | 0.27, 0.74 |
| SCS-SF | -0.143 | -0.342, 0.056 | .157 | -0.16 | -0.40, 0.07 |
| EMAS | 0.281 | -1.425, 1.987 | .746 | 0.04 | -0.20, 0.27 |
| LOT-R | -0.643 | -1.893, 0.607 | .312 | -0.12 | -0.36, 0.11 |
| MSPSS | -2.787 | -7.042, 1.467 | .198 | -0.16 | -0.40, 0.08 |
| ITQ | 3.003 | 1.184, 4.822 | .001 | 0.42 | 0.17, 0.66 |

Stigmatisation Questionnaire (**PSQ**; p < .001, g = 0.51) and the International Trauma Questionnaire (**ITQ**; p = .001, g = 0.42).

Table 4 presents Pearson correlation coefficients for each predictor-outcome combination in both samples. For the veteran sample, all correlation coefficients exceed the critical values for significance at the 5% level (critical value: .174, n = 120, two-sided), the 1% level (critical value: .228, two-sided), and are significant at the 0.1% level (critical value: .281, two-sided)

**Table 4. Pearson correlation coefficients for outcome variables with explanatory variables.**

| | Veteran | | | | | |
|---|---|---|---|---|---|---|
| | BESAA-AE | BILEQ | FNAES | SAS-new | SAS-general | PHQ-9 |
| BIAAQ | .732 | .791 | -.791 | -.598 | -.580 | -.575 |
| BICSI-AF | -.449 | -.422 | .662 | .461 | .329 | .230 |
| PSQ | -.425 | -.456 | .395 | .354 | .429 | .332 |
| SCS-SF | .599 | .616 | -.633 | -.518 | -.566 | -.728 |
| EMAS | .465 | .396 | -.268 | -.311 | -.358 | -.437 |
| LOT-R | .563 | .625 | -.566 | -.526 | -.509 | -.672 |
| MSPSS | .313 | .418 | -.289 | -.238 | -.346 | -.449 |
| ITQ | -.489 | -.655 | .603 | .547 | .516 | .768 |
| | Civilian | | | | | |
| | BESAA-AE | BILEQ | FNAES | SAS-new | SAS-general | PHQ-9 |
| BIAAQ | .698 | .741 | -.716 | -.458 | -.491 | -.601 |
| BICSI-AF | -.570 | -.276 | .779 | .454 | .423 | .366 |
| PSQ | -.438 | -.556 | .501 | .396 | .433 | .466 |
| SCS-SF | .621 | .366 | -.655 | -.645 | -.594 | -.664 |
| EMAS | .468 | .449 | -.415 | -.199 | -.187 | -.500 |
| LOT-R | .533 | .408 | -.524 | -.535 | -.531 | -.547 |
| MSPSS | .255 | .281 | -.298 | -.214 | -.343 | -.301 |
| ITQ | -.508 | -.532 | .574 | .444 | .495 | .628 |

For the veteran sample significance is achieved if: Absolute Correlations > .174 sig (alpha = .05), > .228 sig (alpha = .01), > .289 sig (alpha = .001) two-sided

For the civilian sample significance is achieved if: Absolute Correlations > .159 sig (alpha = .05), > .208 sig (alpha = .01), > .264 sig (alpha = .001) two-sided

except for the correlation between BICSI-AF and PHQ-9 (r = .230). In the civilian sample, all correlation coefficients exceed the critical values for 5% significance (critical value: .159, n = 160, two-sided) and 1% significance (critical value: .208, n = 160, two-sided), except for the correlation between Engagement in Meaningful Activities Scale (EMAS) and Social Anxiety Scale-new situations (SAS-new; r = -.187) and EMAS and Social Anxiety Scale-general situations (SAS-general; r = -.199). Otherwise, all correlation coefficients exceed the critical value for the 0.1% level (critical value: .264, n = 160, two-sided), except for the correlation between Multi-dimensional Scale of Perceived Social Support (MSPSS) and Body Esteem Scale for Adults and Adolescents–Appearance Subscale (BESAA-AE) (r = .255) and the correlation between MSPSS and SAS-new (r = -.214).

## Regression model of factors associated with Body Esteem Scale for Adults and Adolescents–Appearance subscale (BESAA-AE)

The regression model for BESAA-AE for the veteran sample ($R^2$ = .621, p < .001) and the civilian sample ($R^2$ = .628, p < .001) is given in Table 5. In both models there is a significant and positive association between body image psychological flexibility (BIAAQ) and BESAA-AE and between engagement in meaningful activities (EMAS) and BESAA-AE. However, appearance fixing as a coping strategy (BICSI-AF) and self-compassion (SCS-SF) are associated with BESAA-AE in the civilian model only. Conclusions are unchanged after multiple imputation (see S1 Table). The direction of these significant effects aligns with the study hypotheses. The probe "I use humor to defuse awkward conversations about my injury and/or altered appearance" was negatively related to BESAA-AE ($\dot{\beta}$ = -0.128, p = .048) in the veteran sample but was not significant in the civilian sample ($\dot{\beta}$ = -.004, p = .935). The probe "Being physically fit is important to me" made a significant contribution to the model in the civilian sample ($\dot{\beta}$ = .116, p = .029) but not in the veteran sample ($\dot{\beta}$ = .004, p = .955). The probe "I feel disgust when I think about or look at my appearance/body" made a significant contribution to the model for BESAA-AE in both the veteran sample ($\dot{\beta}$ = -.393, p < .001) and the civilian sample ($\dot{\beta}$ = -.517, p < .001). Within the veteran model, the probe "Being seen as an injured veteran is a good thing" made a significant improvement to the modelling of BESAA-AE ($\dot{\beta}$ = +0.173, p = .011) as did the probe for total number of years of military service ($\dot{\beta}$ = -0.142, p = .026).

## Regression model of factors associated with Body Image Life Engagement (BILEQ)

There are significant effects in the BILEQ model for veterans ($R^2$ = .691, p < .001) and civilians ($R^2$ = .608, p < .001) as shown in Table 5. In both models, BIAAQ is significantly associated with BILEQ (p < .001). However, PTSD (ITQ) is negatively associated with BILEQ but only in the veteran sample, whereas BICSI-AF, PSQ, SCS-SF and EMAS are each significantly associated with BILEQ but only in the civilian sample. Conclusions are unchanged after multiple imputation. The direction of the significant effects aligns with the pre-study hypotheses. The probe "I tend to say to myself it could have been worse" (Binary: 0 = No, 1 = Yes) significantly improved the model in the veteran sample ($\dot{\beta}$ = .128, p = .030) but not in the civilian sample ($\dot{\beta}$ = .015, p = .777).

## Regression model of factors associated with fear of negative appearance evaluation (FNAES)

Significant effects for modelling FNAES are observed in the veteran ($R^2$ = .732, p < .001) and civilian sample ($R^2$ = .782, p < .001) as shown in Table 5. In these models, BIAAQ and

**Table 5. Regression models for appearance outcome measures.**

| | BESAA-AE | | | | | |
|---|---|---|---|---|---|---|
| | **Veteran** | | | **Civilian** | | |
| | $R^2 = .621$, p < .001 | | | $R^2 = .628$, p < .001 | | |
| **Measure** | Beta | t | p | Beta | t | p |
| BIAAQ | .593 | 5.828 | < .001 | .485 | 5.869 | < .001 |
| BICSI-AF | -.666 | -0.858 | .393 | -.175 | -2.783 | .006 |
| PSQ | -.002 | -0.029 | .977 | .061 | 0.929 | .355 |
| SCS-SF | .007 | 0.062 | .951 | .186 | 2.403 | .017 |
| EMAS | .254 | 3.423 | .001 | .153 | 2.497 | .014 |
| LOT-R | .054 | 0.511 | .611 | .101 | 1.392 | .166 |
| MSPSS | .006 | 0.086 | .932 | -.044 | -0.803 | .423 |
| ITQ | .003 | 0.036 | .972 | .054 | 0.758 | .450 |
| | **BILEQ** | | | | | |
| | **Veteran** | | | **Civilian** | | |
| | $R^2 = .691$, p < .001 | | | $R^2 = .608$, p < .001 | | |
| | Beta | t | p | Beta | t | p |
| BIAAQ | .552 | 5.986 | < .001 | .658 | 7.695 | < .001 |
| BICSI-AF | -.024 | -0.344 | .732 | .151 | 2.333 | .021 |
| PSQ | -.021 | -0.306 | .760 | -.190 | -2.786 | .006 |
| SCS-SF | .005 | 0.056 | .956 | -.187 | -2.358 | .020 |
| EMAS | .144 | 1.701 | .092 | .188 | 2.981 | .003 |
| LOT-R | .039 | 0.409 | .683 | .093 | 1.243 | .216 |
| MSPSS | .079 | 1.182 | .240 | .003 | 0.058 | .954 |
| ITQ | -.201 | -2.299 | .024 | -.015 | -0.015 | .847 |
| | **FNAES** | | | | | |
| | **Veteran** | | | **Civilian** | | |
| | $R^2 = .732$, p < .001 | | | $R^2 = .782$, p < .001 | | |
| | Beta | t | p | Beta | t | p |
| BIAAQ | -.450 | -5.257 | < .001 | -.330 | -5.211 | < .001 |
| BICSI—AF | .328 | 5.112 | < .001 | .464 | 9.630 | < .001 |
| PSQ | .022 | 0.347 | .730 | .019 | 0.382 | .703 |
| SCS-SF | -.131 | -1.447 | .151 | -.179 | -3.007 | .003 |
| EMAS | -.023 | -0.363 | .718 | -.009 | -0.186 | .853 |
| LOT-R | .038 | 0.429 | .669 | -.078 | -1.402 | .163 |
| MSPSS | .005 | 0.085 | .932 | -.031 | -0.725 | .470 |
| ITQ | .148 | 1.827 | .071 | -.005 | -0.085 | .933 |

BESAA-AE: Body Esteem–Appearance Sub-scale; BILEQ: Body Image Life Disengagement; FNAES: Fear of Negative Appearance Evaluation. BIAAQ: Body Image Acceptance and Action (Body Image Psychological Flexibility; BICSI-AF: Body Image Coping Strategies–Appearance Fixing; PSQ: Perceived Stigma; SCS-SF: Self-Compassion; EMAS: Engagement in Meaningful Activities; LOT-R: Optimism; MSPSS: Multidimensional Perceived Social Support; ITQ: International Trauma Questionnaire (PTSD).

BICSI-AF are both significantly related to FNAES. SCS-SF is also significantly related to FNAES in the civilian sample only. Conclusions are unchanged under multiple imputation. When probing the models, the probe variable "I feel disgust when I think about or look at my appearance/body" was significantly associated with FNAES in the veteran sample ($\dot{\beta}$ = .260, p = .001) and the civilian sample ($\dot{\beta}$ = .172, p = .005).

## Regression model of factors associated with symptoms of depression (PHQ-9)

Table 6 gives the regression model for PHQ-9 with significant effects captured in both the veteran ($R^2$ = .706, p < .001) and civilian samples ($R^2$ = .627, p < .001). Both SCS-SF and ITQ are significantly related to PHQ-9 in both samples, and EMAS is significantly related to PHQ-9 in the civilian sample only. The same statistical conclusions are obtained under multiple

**Table 6. Regression models for factors associated with mental health outcome measures.**

| | **PHQ-9** | | | | | |
| | **Veteran** | | | **Civilian** | | |
| | $R^2$ = .706, p < .001 | | | $R^2$ = .627, p < .001 | | |
| **Measure** | Beta | t | p | Beta | t | p |
| BIAAQ | -.046 | -0.509 | .612 | -.116 | -1.402 | .163 |
| BICSI-AF | -.134 | -1.985 | .050 | -.076 | -1.212 | .227 |
| PSQ | -.010 | -0.147 | .883 | -.009 | -0.132 | .895 |
| SCS-SF | -.229 | -2.398 | .018 | -.340 | -4.383 | < .001 |
| EMAS | -.095 | -1.453 | .149 | -.185 | -3.012 | .003 |
| LOT-R | -.184 | -1.977 | .051 | -.064 | -0.890 | .375 |
| MSPSS | -.088 | -1.354 | .179 | -.057 | -1.026 | .306 |
| ITQ | .442 | 5.189 | < .001 | .334 | 4.646 | < .001 |
| | **SAS-New** | | | | | |
| | **Veteran** | | | **Civilian** | | |
| | $R^2$ = .524, p < .001 | | | $R^2$ = .571, p < .001 | | |
| | Beta | t | p | Beta | t | p |
| BIAAQ | -.201 | -1.760 | .081 | .007 | 0.077 | .939 |
| BICSI-AF | .251 | 2.931 | .004 | .217 | 3.077 | .003 |
| PSQ | .071 | 0.823 | .413 | .074 | 0.980 | .329 |
| SCS-SF | -.252 | -2.082 | .040 | -.428 | -4.821 | < .001 |
| EMAS | -.061 | -0.735 | .464 | .193 | 2.718 | .007 |
| LOT-R | .010 | 0.083 | .934 | -.192 | -2.313 | .022 |
| MSPSS | .081 | 0.979 | .330 | -.026 | -0.404 | .687 |
| ITQ | .179 | 1.648 | .103 | -.086 | 1.048 | .296 |
| | **SAS-General** | | | | | |
| | **Veteran** | | | **Civilian** | | |
| | $R^2$ = .466, p < .001 | | | $R^2$ = .510, p < .001 | | |
| | Beta | t | p | Beta | t | p |
| BIAAQ | -.245 | -2.023 | .046 | .007 | 0.071 | .944 |
| BICSI-AF | .093 | 1.022 | .309 | .192 | 2.656 | .009 |
| PSQ | .121 | 1.325 | .188 | .112 | 1.490 | .138 |
| SCS-SF | -.251 | -1.954 | .054 | -.293 | -3.288 | .001 |
| EMAS | -.103 | -1.168 | .246 | .213 | 3.019 | .003 |
| LOT-R | .055 | 0.439 | .662 | -.201 | -2.418 | .017 |
| MSPSS | -.051 | -0.583 | .561 | -.182 | -2.866 | .005 |
| ITQ | .113 | 0.984 | .327 | .162 | 1.964 | .051 |

PHQ-9: Patient Health Questionnaire (Depression symptoms); SAS-New: Social Anxiety in New Situations; SAS-General: Social Anxiety in General. BIAAQ: Body Image Acceptance and Action (Body Image Psychological Flexibility; BICSI-AF: Body Image Coping Strategies–Appearance Fixing; PSQ: Perceived Stigma; SCS-SF: Self-Compassion; EMAS: Engagement in Meaningful Activities; LOT-R: Optimism; MSPPS: Multidimensional Perceived Social Support; ITQ: International Trauma Questionnaire (PTSD).

imputation (see S2 Table). The direction of the significant effects aligns with the pre-study hypotheses. Probing the PHQ-9 models with non-validated measures did not produce any other statistically significant effects.

## Regression model of factors associated with social anxiety and avoidance in new situations (SAS-new)

For SAS-new there are significant associations in the veteran ($R^2$ = .524, p < .001) and civilian samples ($R^2$ = .571, p < .001) (see Table 6), with BISCI-AF and SCS-SF statistically significant in both models. However, under multiple imputation, the association between SCS-SF and SAS-new is not deemed significant in the veteran sample (see S2 Table). EMAS and optimism (LOT-R) are each significant predictors of SAS-new in the civilian sample but not in the veteran sample. The direction of the significant effects aligns with the pre-study hypotheses. The variable "I feel disgust when I think about or look at my appearance/body" was significantly associated with SAS-new in the civilian sample ($\dot{\beta}$ = .182, p = .043) but not in the veteran model ($\dot{\beta}$ = .025, p = .821). Similarly, the variable "How I think I look inside my own head is the same as how I look to others" was significantly associated with this outcome in the civilian sample ($\dot{\beta}$ = .144, p = .029) but not in the veteran sample ($\dot{\beta}$ = .006, p = .942). When probing the veteran model, "Being seen as an injured veteran is a good thing" significantly improved the SAS-new model ($\dot{\beta}$ = .158, p = .041), as did the variable "Having other veterans/military personnel close by who are recovering from similar injuries and changes to their appearance has been helpful" ($\dot{\beta}$ = .153, p = .046).

## Regression model of factors associated with social anxiety and avoidance in general situations (SAS-general)

As shown in Table 6, there is some evidence of significant associations in the civilian sample ($R^2$ = .510, p < .001) with BISCI-AF, SCS-SF, EMAS, LOT-R, and social support (MSPSS) all related to SAS-general. However, the evidence of significant associations between measures and SAS-general in the veteran sample ($R^2$ = .466, p < .001) is less compelling because the BIAAQ is deemed significant when working with complete cases but not under multiple imputation noting p = 0.06 (see S2 Table). When probing the model, the variable "I feel disgust when I think about or look at my appearance/body" was significantly associated with this outcome in the civilian sample ($\dot{\beta}$ = .187, p = .037) but not in the veteran model ($\dot{\beta}$ = .041, p = .727).

## Discussion

This study aimed to fill a gap in understanding the psychosocial support needs of veterans with conflict-related AAI. The goal was to identify factors predicting psychosocial adjustment among veterans compared to civilians without a military background. The study focused on outcomes indicative of adjustment, such as body esteem, body image (BI) life engagement, fear of negative appearance evaluation, social anxiety and avoidance, and depression. Potential predictors were informed by existing research on civilians with visible differences and insights from a preceding qualitative investigation with injured veterans [18].

Correlations between predictors and outcomes were all significant and the direction of relationships were as hypothesised, and similar between groups. Across both groups, BI psychological flexibility, self-compassion, engagement in meaningful activities, optimism and social support were positively and significantly correlated with body esteem and life engagement, and negatively and significantly correlated with fear of negative appearance evaluation, social anxiety, and depression. Across both groups, appearance fixing, perceived stigma and PTSD

were negatively and significantly correlated with body esteem and life engagement, and positively and significantly correlated with fear of negative appearance evaluation, social anxiety, and depression. These findings, combined with evidence from regression analyses that our predictor variables explained a high degree of variance in the outcomes, support the conceptualisation of adjustment proposed from the outset, and thus improve understanding of the key factors that exacerbate and ameliorate appearance-related distress among those with a visible difference.

Overall, civilians and veterans reported similar experiences of living with an AAI, with some key exceptions. The strength of correlations between self-compassion and BI life engagement, and PTSD and depression, were significantly stronger among veterans. Veterans also experienced significantly lower BI psychological flexibility, were more likely to avoid social, recreational, and vocational activities due to appearance concerns (low BI life engagement), experienced significantly greater depression and PTSD, and perceived more appearance-related stigmatising behaviours by others.

Research suggests a higher prevalence of mental health difficulties, including PTSD, among UK veterans compared to the public [36]. The reasons for these differences in mental health outcomes remain unclear, although pre-service vulnerabilities [36], trauma specific to military conflict [37], experiences of transition out of the military [38], and poor help-seeking behaviour [39], are acknowledged risk factors. While non-appearance related factors likely contribute to the increased incidence of PTSD and depression among veterans, this study's findings reveal worrying differences in appearance-related constructs, suggesting they may be more vulnerable to the impact of an AAI. Despite similarities in body esteem and fear of negative appearance evaluation, increased tendency for lower BI flexibility, heightened perception of stigmatising behaviour by others, and reduced life engagement, are outcomes that signify poor coping relative to similar others in the general population and may account for the stronger relationship between PTSD and depression among veterans. Additionally, indication that self-compassion may help veterans overcome appearance concerns related to social activities (BI life engagement) provides preliminary evidence that self-compassion may be a beneficial target via intervention.

Multiple regression analyses determined that **BI psychological flexibility** played the most significant role in predicting adjustment among both groups. It was strongly associated with all appearance-related outcomes, predicting higher body esteem and BI life engagement and lower fear of negative appearance evaluation. Lower BI psychological flexibility was also a significant, although not robust, predictor of heightened social anxiety and avoidance in general situations among veterans.

A recent meta-analysis supports the positive role of BI psychological flexibility in adaptive processes related to body related and mental health indices [40]. BI psychological flexibility was consistently negatively correlated with constructs indicative of body image concerns, depression, anxiety, and general psychological distress, and positively associated with positive body-related constructs, including body appreciation and body acceptance. Shepherd et al. [11] reported that psychological *in*flexibility (the negative form of psychological flexibility) was positively associated with appearance anxiety among individuals with a visible burn injury. A study involving individuals with various appearance-altering conditions [12] found that cognitive fusion and experiential avoidance (negatively valenced components of psychological flexibility) partially mediated the relationship between body esteem and appearance fixing (a coping strategy), measured using the BILEQ. Experiential avoidance also partially mediated the relationship between body esteem and behavioral avoidance. Consequently, activities promoting cognitive *defusion* and experiential acceptance may benefit individuals with visible differences.

Prior research and our findings suggest that BI psychological flexibility may lessen appearance-related distress in civilians and veterans with AAI. Individuals who embrace body image threats with kindness and acceptance, rather than resorting to unhelpful strategies like social avoidance, are more likely to thrive. Acceptance and Commitment Therapy (ACT), a transdiagnostic third-wave cognitive-behavioral therapy and behavior change model [41], employs techniques to foster psychological flexibility, aiming to help individuals lead a more fulfilling and meaningful life. These include developing mindfulness to de-identify from thoughts (cognitive defusion) and open-up to painful emotions (experiential acceptance) and helping individuals to clarify their values to inform their actions via goal setting (committed action). Psychologists across Europe have noted the utility of ACT for patients with visible differences [42, 43], and there is some empirical research of its effectiveness and suitability in new interventions [44, 45].

In both groups, **engagement in meaningful activities** predicted higher body esteem, which is congruent with findings related to BI psychological flexibility; engaging in meaningful activities consistent with one's values and needs arguably equates to the process of committed action (acting towards goals guided by values) that fosters psychological flexibility [46]. However, engagement in meaningful activities only predicted greater BI life engagement and higher social anxiety among civilians. These significant relationships may exemplify psychological flexibility where individuals motivated by pursuing values, engage in socially exposing activities despite increased social anxiety. Absence of this finding in veterans may be related to indications of lower BI psychological flexibility, perceived appearance-related stigma, and increased avoidance of social activities due to appearance concerns. Keeling et al. [18] qualitative research with 23 military participants who sustained AAI also details the negative impact of intrusive appearance-related social stigma (e.g., being stared at, being insulted), particularly a depleted sense of social anonymity.

ACT-based interventions could benefit both groups, especially veterans. Veterans may also benefit from social skills training aimed at increasing confidence in managing challenging public situations. Civilians with visible differences have found this approach helpful [47–49].

**Self-compassion** also distinguished veterans from civilians. Lower self-compassion predicted depression and social anxiety in both groups, consistent with a meta-analysis [48]. Only among civilians did lower self-compassion predict increased fear of appearance evaluation, while higher self-compassion predicted elevated body esteem. Prior research indicates that veterans may actively resist self-compassion [49, 50]. Distinct from low self-compassion, 'Fear of compassion' [50] is associated with feeling undeserving, viewing compassion as a weakness that might expose flaws or infer lowered personal standards, or simply not appreciating its value. Fear of compassion might explain differences between civilians and veterans, with the latter more likely to endorse military values such as courage, stoicism, and collectivism [51].

Both psychological flexibility and self-compassion reflect a common core of mindful (i.e., open, non-judgmental) awareness concerning emotional distress [52], and processes targeted via ACT are inherently self-compassionate. Steen et al. [53] highlighted the benefits of focusing on self-compassion for veterans, especially those with combat experience, trauma, or PTSD. Our findings suggest that veterans with AAI could benefit from targeted self-compassion interventions.

Civilians showed a relatively greater involvement in **appearance fixing**, concealing worrisome aspects, or seeking reassurance about appearance. In both groups, engaging in these behaviors predicted fear of negative appearance evaluation and social anxiety in new situations (e.g., meeting people for the first time). However, only among civilians did increased engagement in appearance fixing predict lower body esteem, higher life engagement related to body image, and heightened social anxiety in general situations (e.g., among peers). These findings

highlight the nuanced relationship between appearance fixing and psychosocial wellbeing. Cash [28] identified appearance-fixing tendencies, like body-concealment using clothing, wigs, prosthetics, or extensive makeup, as an avoidant coping strategy linked to body dissatisfaction and sustained social anxiety. However, concealment and seeking reassurance may also provide enough social confidence to encourage participation in appearance-oriented activities that would otherwise be avoided [54]. Our evidence suggests veterans are generally more inclined than civilians to avoid social activities due to appearance concerns (BI life engagement). Specifically, while veterans engage in appearance fixing when anxious about new social situations, they do not appear to experience the same benefits as civilians with low body esteem and high social anxiety, who use appearance fixing to facilitate engagement in general social, recreational, and vocational activities (BI life engagement). This could be attributed to veterans' reluctance to engage in behaviors perceived as vain or threatening masculinity, as indicated by Keeling et al. [18]. This needs consideration when designing veteran-specific interventions, but further exploration is necessary to confirm and understand these differences.

Previous research indicates the benefits of dispositional optimism and engaging social support in buffering stress effects for civilians with visible differences, especially using social support as a coping strategy during exposure to feared social situations [2]. Among civilians, our findings align, with lower optimism associated with increased social anxiety in new situations and lower social support associated with heightened social anxiety in general situations. However, this was not observed among the veterans. This was unexpected, given that optimism has been recognized as a buffer protecting veterans exposed to combat stress from mental health symptoms such as depression and PTSD [55] and, more specific to appearance, findings from Keeling et al. [18] where injured veterans discussed the value of positive reframing (i.e., their injuries could have been fatal or worse), adaptive coping and accepting what they cannot change. In the same qualitative study [18], veterans emphasized the benefits of social support, particularly the camaraderie during rehabilitation. Our findings suggest nuanced differences in beliefs among veterans regarding their military experiences and their perspectives on appropriate support. Notably, '**believing their injuries could have been worse**' appeared to promote greater BI life engagement, possibly reflecting the use of downward social comparison as a coping strategy, which was seemingly more pertinent to those who had frequently witnessed life-changing injuries among, or the death of, comrades. Similarly, the finding that veterans' belief that '**it is important to recover alongside injured veterans'** increased social anxiety in new situations suggests that some veterans value or need support from those with shared experiences to facilitate social engagement. Alternatively, some predictors might not consistently be significant in the regressions due to high overall R-squared values for the field. This could make it challenging for another predictor to contribute when key predictors already account for much of the variance. This might also explain why lower perceived stigma was only associated with higher BI life engagement among civilians, as veterans overall experienced greater perceived stigmatization and were more likely to avoid appearance-related social activities.

Of the variables measured using questions created by the authors, heightened feelings of **self- disgust** predicted lower body esteem and higher fear of negative evaluation for both groups. Previous research indicates that physical self-disgust, a visceral revulsion toward oneself, akin to self-stigma and shame, can arise when individuals perceive their physical appearance violates societal norms. It is associated with higher body image dissatisfaction in conditions like limb amputation [56] and has been implicated as a mediating factor between BID and suicidal ideation among a large non-clinical sample [57]. While only drawing evidence from a single item rather than a validated scale, our findings suggest both veterans and civilians with AAI may benefit from interventions targeting self-disgust. Powell, Simpson, and Overton [58] found that self-affirmation techniques emphasising non-appearance-related

traits to bolster self-worth, with a focus on kindness, reduced appearance-directed disgust. Evidence also indicates the utility of ACT, with a focus on self-compassion, for addressing self-stigma and shame [59].

Other factors influencing body esteem among veterans included perceiving it is **'a good thing to be viewed as a veteran by the public'**, predicting higher body esteem, while **using humor** to diffuse awkward social situations predicted lower body esteem. The former may indicate veterans taking pride in their service and seeking recognition, potentially guarding against misperceptions or emasculation judgments related to their injuries. This aligns with values reinforced through military acculturation [51]. The use of 'dark humor' as an adaptive strategy to cope with stress, common in the military [60], may promote morale and protect against PTSD [61]. Interventions for civilians often recommend the use of humor to manage challenging social interactions; to appear confident and put others at ease who appear nervous or unsure how to respond to their visible difference [62]. Our findings advise interventionists to be cautious in assuming that veterans who rely on humor are confident; humor may also reflect vulnerability (low body esteem) and the need for additional coping strategies.

Unsurprisingly, PTSD symptomology was associated with depression across groups, a common finding [63], but it specifically impacted adjustment by reducing BI life engagement and increasing fear of negative appearance evaluation among veterans. The higher incidence of PTSD among veterans and its comparatively greater role in appearance-related constructs suggest that interventions for appearance-related distress among veterans should incorporate a trauma-focused approach which recognizes the widespread impact and signs of trauma, aiming to prevent re-traumatization.

## Strengths and limitations

This study, one of the first to compare the experiences of military veterans and civilians with appearance-altering injuries, highlights key differences between the groups which raise important implications for the design and delivery of psychological interventions. All regression models had extremely high overall R-squared values and therefore excellent goodness of fit.

Limitations include the cross-sectional design, meaning causation cannot be inferred. The self-selecting recruitment approach has implications for generalisability, potentially leading to underreported concerns or exclusion of those with significant issues. French et al. [64] reported that facial and limb injuries were significant predictors of posttraumatic stress, yet hypothesized that military service members may under-report symptoms, suggesting that the impact of bodily injuries could be greater than the limited evidence indicates. Generalisability is further limited by the small proportion of females and those of ethnicities other than white. Additionally, there is evidence to suggest that men are more likely to under-report negative experiences on questionnaires [65] which may also have affected the results. Despite the models explaining a high level of variance, unmeasured factors might have influenced the outcomes. This may be especially true for participant age and injury cause. These factors may be particularly pertinent for veterans where era of Service and whether an injury was sustained during combat or not, may affect veterans' perceptions and adjustment. Finally, it is noted that this research focuses on UK military veterans. Due to differences in cultural appearance norms, health care systems and other military related nuances, generalizability to veterans of other country's militaries should be conducted cautiously.

## Conclusion

This novel research confirms the multivariate nature of psychosocial adjustment to an AAI among civilians and veterans, adding to and further supporting existing evidence. It highlights

key similarities and differences between these groups, indicating that veterans may be more vulnerable to the psychosocial impact of an altered appearance. Key factors associated with adjustment include BI psychological flexibility, engagement in meaningful activities, and self-compassion. This provides evidence that ACT-based interventions that include explicit self-compassion activities, as well as social skills training for managing difficult social situations, may benefit both groups, but particularly veterans. In addition, this study provides evidence of veteran-specific differences such as the potential benefit of a perceived positive regard for injured veterans by the public, the use of humor to manage difficult situations, beliefs about injury, appearance, and the nature of appropriate support, and heightened PTSD symptoms. These military specific factors should be considered in the development and provision of interventions, including ensuring a trauma-informed approach is taken.

## Supporting information

**S1 Table. Pooled multiple imputation regression models.**
(DOCX)

**S2 Table. Pooled multiple imputation regression models for PTSD and social anxiety.**
(DOCX)

## Acknowledgments

The authors wish to thank the participants and the study Public and Patient Involvement advisors for their time and sharing their experiences, the study steering committee for their time and contributions, and all the support services, veteran organisations, NHS services, and individuals, who supported participant recruitment.

## Author Contributions

**Conceptualization:** Mary Keeling, Diana Harcourt, James Kiff, Heidi Williamson.

**Data curation:** Mary Keeling, Paul White, Victoria S. Williams V., Heidi Williamson.

**Formal analysis:** Mary Keeling, Diana Harcourt, Paul White, Heidi Williamson.

**Funding acquisition:** Diana Harcourt, Heidi Williamson.

**Methodology:** Mary Keeling, Diana Harcourt, Paul White, James Kiff, Heidi Williamson.

**Project administration:** Mary Keeling, Sarah Evans, Victoria S. Williams V.

**Supervision:** Diana Harcourt.

**Writing – original draft:** Mary Keeling.

**Writing – review & editing:** Diana Harcourt, Paul White, Sarah Evans, Victoria S. Williams V., James Kiff, Heidi Williamson.

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
