## [Decision Letter · Decision Letter 0]

9 Sep 2024

PONE-D-24-20100Body image and appearance distress among military veterans and civilians with an injury-related visible difference: A comparison studyPLOS ONE

Dear Dr. Keeling,

Thank you for submitting your manuscript to PLOS ONE. After careful consideration, we feel that it has merit but does not fully meet PLOS ONE’s publication criteria as it currently stands. Therefore, we invite you to submit a revised version of the manuscript that addresses the points raised during the review process.

We look forward to receiving your revised manuscript.

Kind regards,

Remington Nevin, MD, MPH, DrPH

Academic Editor

PLOS ONE

Journal Requirements:

"NO authors have competing interests"

Additional Editor Comments:

Thank you for your patience as appropriate and qualified reviewers were secured. Your manuscript has now been reviewed by two reviewers, both of whom responded favorably.

Please note Reviewer #1’s comments regarding standardizing citation formatting. Please consult the journal’s instructions to authors for further guidance. Reviewer #1 also raises several important questions, each of which should be addressed either in the text, or in a response to the reviewer.

Reviewer #2 raises important concerns regarding the heterogeneity of veteran subjects’ wartime experiences and perspectives and thus how these may influence certain outcomes of interest in this study. Based on these concerns, it appears these may correlate with PTSD (and hence ITQ) and could partially account for some of the study’s findings. These concerns and this possibility should be addressed in the text in your revision.

Line 45: The “injury, wounds, and scarring” compensation figures likely include significant numbers of musculoskeletal injuries and may therefore be relatively non-specific for the significant AAIs of interest in this context. This should be clarified or a more specific figure cited.

Reviewers' comments:

Reviewer's Responses to Questions

**Comments to the Author**

1. Is the manuscript technically sound, and do the data support the conclusions?

Reviewer #1: Yes

Reviewer #2: Partly

2. Has the statistical analysis been performed appropriately and rigorously? 

Reviewer #1: Yes

Reviewer #2: Yes

3. Have the authors made all data underlying the findings in their manuscript fully available?

Reviewer #1: Yes

Reviewer #2: Yes

4. Is the manuscript presented in an intelligible fashion and written in standard English?

Reviewer #1: Yes

Reviewer #2: Yes

5. Review Comments to the Author

Reviewer #1: Thank you for submitting this important paper highlighting body image and appearance distress of this under-studied population.

However, I do have some questions and remarks.

• What surprises me in the text is that numbers and names are referred to alternately. Perhaps it is the journal's policy but it does not read pleasantly. For example, from line 59 to line 63: reference is made to (6) and to Shepherd, 2015.

• The abstract mentions that ACT is the most appropriate intervention because of self-compassion and social skills training. I find this conclusion a bit ‘over-simplified’: there are also other interventions that focus on self-compassion and skills training. Can the authors better explain in 1 or more sentences why ACT is most appropriate in this one?

• The introduction mentions that symptoms of PTSD are negatively associated with body esteem and body life engagement. I wonder if PTSD belongs in this list? and to what extent you can include this in further research? PTSD can obviously have a far-reaching effect on well-being but is one of a different category than e.g. body image or perceived social support: PTSD is a formal psychiatric diagnosis. Can the authors indicate to what extent they think PTSD belongs here?

• It is rightly indicated under the limitations of the study that the number of women in the group of veterans was limited. I also miss the argument here that generally men under-report on questionnaires.

• In addition, the variation in years since injury is considerable: among veterans, the average is 18 years. How does that affect the perception of the impact of the injury? Do the authors have an idea about that?

Reviewer #2: My study focused on USA female veterans between the ages of 24 and 42. Seven lost one or more limbs in combat. One lost a lower limb stateside but served a tour of duty on a prosthesis in Iraq. I observed that these veterans viewed their loss of limb(s) as a mark of valor and eschewed cosmetic prosthetics.

Since loss of limb affected their ability as soldiers, all were forced to reassess their career and life expectations. Using a group of veterans that included vets aged 28 to 75 with a median age of 42 would have influenced the data. One of my vets was very active with veterans organization throughout the USA and she confirmed my observation that Iraqi vets attitudes towards their limb loss differed from prior veteran cohorts. Mixing individuals who were injured in combat with those injured through accidents also may have skewed the data due to differing perceptions about their injuries and its affect on their body image. In my experience as a vocational rehabilitation counselor those veterans who were injured in basic training had a much different attitude versus those injured in combat. Also attitudes between American vets and UK may differ as acknowledged by authors.

I appreciate the in-depth study and believe it will be useful to those entrusted with the care of these injured vets as long as counselors keep in mind that different veteran cohorts may have different body image issues. The rehabilitation received by older veteran age cohorts may differ greatly from the rehabilitation process in this era. Also the cultures in which one sustained serious injury may influence body image significantly.

6. PLOS authors have the option to publish the peer review history of their article (what does this mean?). If published, this will include your full peer review and any attached files.

Reviewer #1: No

Reviewer #2: **Yes: **Janet K Cater

---

## [Editor Report · Decision Letter 1]

6 Nov 2024

Body image and appearance distress among military veterans and civilians with an injury-related visible difference: A comparison study

PONE-D-24-20100R1

Dear Dr. Keeling,

We’re pleased to inform you that your manuscript has been judged scientifically suitable for publication and will be formally accepted for publication once it meets all outstanding technical requirements.

Kind regards,

Remington Nevin, MD, MPH, DrPH

Academic Editor

PLOS ONE

---

## [Editor Report · Acceptance letter]

15 Nov 2024

PONE-D-24-20100R1 

PLOS ONE

Dear Dr. Keeling, 

I'm pleased to inform you that your manuscript has been deemed suitable for publication in PLOS ONE. Congratulations! Your manuscript is now being handed over to our production team.

Kind regards, 

on behalf of

Dr. Remington Nevin 

Academic Editor

PLOS ONE